# BEYOND STATIC RETRIEVAL POLICIES: TASK-AWARE ADAPTIVE RAG WITH METAR

## ABSTRACT

Large Language Models (LLMs) accompanied by retrieval augmented generation (RAG) have been widely applied to knowledge-intensive tasks due to their strong generalization and contextual understanding capabilities. However, indiscriminate use of RAG can increase computational overhead and degrade performance. Adaptive RAG (ARAG), which dynamically determines whether to retrieve, has emerged as a promising solution. In current literature, ARAG methods typically rely on static decision policies, such as fixed confidence thresholds or task-specific prompts, which are brittle and lack adaptability to diverse task domains. Such task-brittleness leads to significant performance degradation when encountering an unseen task, hindering the real-world applicability of these methods. In this work, we formally define the problem of task adaptability for ARAG and introduce quantitative metrics to benchmark the current methods. To improve task adaptability, we propose METAR (**M**emory-**E**volving **T**ask-**A**ware **R**AG), a novel agentic framework where an agent learns and refines a procedural memory of task-specific retrieval criteria. Experiments across a wide range of tasks show that our method achieves superior task adaptability compared to existing ARAG approaches.

## 1 INTRODUCTION

Large Language Models (LLMs) have demonstrated their power over a wide variety of tasks in natural language processing (Zhao et al., 2023). However, in knowledge-intensive tasks where background knowledge is required for models to perform well, the factuality of LLMs is still a problem. LLMs suffer from hallucinations (Huang et al., 2025), a widely observed phenomenon where LLMs generate incorrect or even nonsense content, due to the incompleteness of their parametric knowledge. Retrieval-Augmented Generation (RAG) comes as a solution, enabling the LLMs to combine knowledge from external resources with their own parametric knowledge, thereby alleviating the problem of hallucination (Fan et al., 2024). However, not all queries to LLMs require external knowledge. Some queries do not need knowledge at all, while some queries can be independently solved by LLMs themselves. In fact, applying RAG indiscriminately leads to irrelevant or wrong information retrieval, which not only increases the computational overhead but also damages factuality (Gao et al., 2024). In the current literature, adaptive RAG (ARAG), also termed active RAG and dynamic RAG, is proposed to dynamically determine whether to retrieve external information to aid in responding to the query (Sharma, 2025).

In this work, we present the task-adaptability problem of ARAG methods. Among current literature of adaptive retrieval, most existing works (Zhang et al., 2024; Su et al., 2024; Ding et al., 2024; Yao et al., 2024; Cheng et al., 2024; Jiang et al., 2023) are dedicated to optimize their performance over a specific question-answering (QA) task, e.g., multi-hop QA and time-sensitive QA. Their adaptability when applied to new tasks remains untested. To be specific, we ask the following question: *Can an ARAG method optimized for a particular task be applied to other tasks while maintaining comparable performance?*

To understand why authentic task adaptability is hard, we propose *the Task-Dependent Knowledge Structure hypothesis* (section 3.2), which attributes the difficulty of task adaptable retrieval decision to the fact that LLM knowledge representations are heterogeneous. Our empirical validations further illustrates that the language model itself relies on linguistic biases to make retrieval decisions, while the existing ARAG methods mostly utilizes brittle latent features that is hard to generalize.

Given these observations, we introduce **M**emory-**E**volving **T**ask-**A**ware **R**AG (METAR), a novel agentic framework for task adaptable retrieval decision. Our framework exploits a procedural memory module to store task-specific solutions. These retrieval criteria can be continuously refined by a reflection model that examines the retrieval decisions and proposes modifications to the solutions. This framework is training-free and requires minimal supervision, making it highly practical and compute-efficient for deployment in real-world LLM+RAG question-answering systems. Our framework operates entirely within the natural language space, without interacting with model parameters, hidden states or output probabilities. This sole reliance on natural language not only enables the framework to be applied to both black-box and white-box LLMs, but also provides users with a convenient way to supervise and intervene in its operation mechanisms.

We evaluate the effectiveness and task-adaptability of our framework across a diverse range of tasks and compare it against multiple baselines. Compared to existing ARAG methods that make use of prompt engineering, probing classifiers and confidence thresholds, our framework outperforms these methods in most tasks and demonstrates decent adaptability to diverse task domains, including even unseen ones. These results suggest that our framework offers significant advantages in real-world applications where LLMs encounter various question answering tasks.

Our main contributions are as follows: (1) We are the first to identify and formally define the task adaptability problem in ARAG methods, a critical yet previously overlooked factor for the performance of practical LLM+RAG question-answering systems; (2) We benchmark a range of existing ARAG methods across a wide variety of tasks, quantitatively assessing their task adaptability; (3) We propose METAR, a novel agentic framework in ARAG settings, which leverages memory-based task-specific solution storage and reflection-driven refinement to achieve task adaptability; (4) We conduct experiments to validate that our framework ourperforms existing ARAG methods in task adaptability.

## 2 RELATED WORKS

### 2.1 ADAPTIVE RETRIEVAL-AUGMENTED GENERATION

As is discussed above, adaptive retrieval-augmented generation (ARAG) allows the LLM to dynamically determine when and what to retrieve so that the efficiency and effectiveness of RAG module is increased. Currently, ARAG methods can mainly be categorized into three types: prompt-based, probing-based or uncertainty-based, while control token-based methods (Asai et al. (2023)) stand out as a different line of research that somehow resembles agentic workflows.

**Prompt-based ARAG** The representative work is RetrievalQA (Zhang et al. (2024)), which creates a new dataset from popular question-answering datasets and designs a prompt to handle retrieval decisions for this new dataset. Besides naive prompt designs, KnowledgeCard (Feng et al. (2023)) trains small task-specific models called knowledge cards as external knowledge resources. The LLM directly determines which knowledge card to use for a given query. These works are relatively simple to implement, but imposes high requirements on the language model itself to make retrieval decisions end-to-end.

**Probing- and Uncertainty-based ARAG** These type of methods probes into the model's outputs, hidden states or parameters and extract their features to determine whether external knowledge is required. ROWEN (Ding et al. (2024)) proposes to paraphase these queries into semantically similar questions and do retrieval when a consistency score falls below a threshold. In addition to consistency thresholds, CBDR (Jin et al., 2025) uses the confidence score for thresholding, while MIND (Ji et al., 2025) calculates token entropy and probes into the attention signals of LLMs. DRAGIN (Su et al. (2024)) probes the hidden states and train a neural network for retrieval decisions. Similarly, SIM-RAG (Yang et al., 2025) also train a neural network for retrieval decisions, but especially focused on multi-round RAG. UAR (Cheng et al. (2024)) points out that single-criterion retrieval judgement isn't enough, and trains multiple classifiers corresponding to multiple criteria. There is also investigation put emphasis on the self-awareness (SeaKR, Yao et al. (2024)), probing the hidden states for the EOS token to predict uncertainty. These method generally requires heavy computation to train a classifier or massive tuning of thresholds to fit the data sampled from a single dataset.

Besides, there have been works trying to understand what kind of methods is better in ARAG settings. However, existings works have reached inconsistent conclusions in answering this question. For example, Soudani et al. (2025) claims that uncertainty estimation falls short, while Moskvoretskii et al. (2025) suggests the exact opposite. Moreover, these works have mostly focused on superficial analysis of the performance of ARAG methods, lacking a deeper insight into the mechanism of how retrieval decisions are made in adaptive RAG. Therefore, we propose to understand the existing works from the perspective of interpretability and try to promote deeper understanding of existing ARAG methods.

## 2.2 LLM Agents

Concurrent AI agents feature several capabilities: perception, understanding and reasoning, planning and acting, as well as reflection. The agentic frameworks generally involve components like tool use, memory management and task planning to equip the LLM-based system with the above skills. ReAct (Yao et al. (2023)) proposes alternating reasoning and acting, which significantly increases the stability and transparency. AutoGPT (Significant Gravitas (2023)) designs a goal-oriented autonomous agent that can decompose tasks, retrieve external resources and execute continuously. Later works like Toolformer (Schick et al. (2023)), ToolChain (Zhuang et al. (2024)) has focused on tool-use and are targeted to improve the LLMs' abilities in real-world scenarios.

Among the recent works in this field, we have drawn inspiration from procedural memory, which is defined as storage of "how-to" knowledge such as how to perform a new skill. AutoManual (Chen et al., 2024) and Memp̂ (Fang et al., 2025) can be seen as one of the pioneering explorations in this new type of agent memory, focusing on LLM planning tasks in complex environments that can benefit from procedural memory through memorizing the understandings of the current environments and rules. Similarly, our ARAG setting also features the requirement for LLM abilities to handle external information and make according decisions. Therefore, we adopt the idea of procedural memory and build our own procedural memory system tailored for ARAG settings.

## 3 The Task-Adaptability Problem of ARAG

In this section, we introduce the task adaptability problem in adaptive RAG settings. First, we define task adaptability in retrieval decision-making (section 3.1). Then we propose our core hypothesis (section 3.2), attributing the difficulty of task adaptability to the fact that LLMs encode knowledge of different domains differently. Finally, some experimental results are provided as empirical evidence of the hypothesis (section 3.3).

### 3.1 Definition of Task-Adaptability in ARAG

In an adaptive RAG situation, the language model takes an input query token sequence $X = [x_1, x_2, ..., x_n]$ which is sampled from the distribution $\mathcal{X}(T)$ of some certain task $T$. It then determines whether to retrieve (indicated by $J(X) \in \{0, 1\}$) and what to retrieve (the retrieval keywords $s(X)$). Therefore, from this perspective, the RAG module is considered an external tool that can be dynamically invoked, instead of an intrinsic module in the question-answering pipeline. Our work on adaptive RAG is considered a study of the language model's ability to fully utilize the RAG module as an external tool, rather than a study of the RAG system itself.

Given a set of tasks $\mathcal{T} = \{T_1, ..., T_k\}$, we define the task adaptability of an ARAG method across these tasks. We assume a performance measure $P_{T_i}(J, \bar{J})$ for each task $T_i \in \mathcal{T}$ which takes the retrieval decision function $J$ of an ARAG method as the input, and compares them against the actual necessity of retrieval $\bar{J}$. This actual necessity is not an inherent property of the query itself, but instead relies on the language model's knowledge capacity. We assess the actual necessity by asking the LLM to provide an answer for $k$ times, and assume it does not retrieve if the percentage of correct answers is above a certain threshold.

$$P_{T_i}(J, \bar{J}) = f_i\Big(\big\{ \big( J(X), \bar{J}(X) \big) \,|\, X \sim \mathcal{X}(T_i)\big\}\Big). \tag{1}$$

Here $f_i \in (\{0, 1\} \times \{0, 1\})^m \to [-1, 1]$ (where $m$ denotes the number of samples from task $T_i$) is a function that provides a scalar performance measure. The function value 1 indicates perfect matching; 0 corresponds to random guessing; $-1$ means completely incorrect predictions.

Practically, the ARAG method for retrieval decision $J$ is usually optimized for a subset of training tasks $\mathcal{T}_{\text{train}} \subset \mathcal{T}$. To evaluate the adaptability of the ARAG method, we assess its performance on the remaining, unconsidered tasks $\mathcal{T}_{\text{remain}} = \mathcal{T} - \mathcal{T}_{\text{train}}$ and quantify the expected performance and the worst-case performance in the remaining tasks as is shown in equation 2.

$$P_{\mathcal{T}_{\text{remain}}}^{\text{avg}} = \mathbb{E}_{T \in \mathcal{T}_{\text{remain}}}[P_T], P_{\mathcal{T}_{\text{remain}}}^{\text{worst}} = \min_{T \in \mathcal{T}_{\text{remain}}} P_T. \tag{2}$$

## 3.2 THE TASK-DEPENDENT KNOWLEDGE STRUCTURE HYPOTHESIS

Retrieval decision-making lies at the heart of ARAG systems, preventing the LLM from processing unnecessary or wrong context while ensuring that relevant information is taken into account. The challenge is that retrieval decisions require accurate and efficient *self-knowledge assessment*. In other words, the LLM must effectively ask, "Is my parametric knowledge sufficient to answer the query?" We hypothesize that the primary obstacle to task adaptability is that the criteria for assessment varies across different contexts.

Inspired by Yu et al. (2023) and Sid Black (2022), which, from a mechanistic interpretability standpoint, suggest that the factual recall mechanism in LLMs differ across contexts, we put forward our central hypothesis. The hypothesis, termed *the Task-Dependent Knowledge Strcture Hypothesis*, is as follows:

**Hypothesis 1.** *The LLMs' internal representation of knowledge is heterogeneous. The way in which an LLM encodes knowledge vary significantly across disciplines, domains and tasks.*

A natural implication is that, a static retrieval policy optimized for one task (such as a fixed confidence threshold, a universal prompt and a statically trained classifier) would be brittle. Such a policy implicitly exploits the knowledge representation features of its training domain that do not transfer to other tasks well.

For intuitive understanding, consider a binary retrieval classifier that is trained in the medical domain by probing into the LLM's attention mechanism. The medical knowledge might be concentrated in several specific attention heads, so a classifier utilizing these features likely performs well in medical fields. However, the same features may provide little to no benefit in other fields like mathematics or physics. Therefore, to build a task-adaptable ARAG system, we want the retrieval policies conditioned on the nature of the task itself.

## 3.3 EMPIRICAL STUDIES: WHY STATIC RETRIEVAL POLICIES FAIL?

In this section we present a series of empirical studies to support and validate the proposed *Task-Dependent Knowledge Structure Hypothesis*. Specifically, we conduct analyses across two representative approaches that covert the major paradigms currently used in adaptive retrieval: prompting-based methods and probing-based binary classifiers. In these studies, we try our best to illustrate how existing methods work and how they might fail when encountering different tasks. For more results and technical details of the following experiments, please refer to the appendix B.

**Experiment I: prompting methods.** To investigate how retrieval decisions are made by LLMs in prompting-based methods, we examine two key components: **the instruction** and **the query**. We randomly sample 100 instances from the NaturalQA (Kwiatkowski et al., 2019) dataset (50 answerable by the LLM, 50 unanswerable). For instructions, we first measure the distribution of retrieval rates under six near-equivalent instructions and RetrievalQA (Zhang et al., 2024), a prompting method proposed for new-world and long-tail knowledge. For queries, we attribute the retrieval decision to token groups using Shapley values (Shapley, 1951) under the same instruction. Specifically, we categorize the query tokens into keywords (content-bearing tokens) and non-keywords (semantic connectives or other linking words) and measure their Shapley values respectively.

Figure 1 shows that even minimal changes in instructions drastically change the distribution of retrieval rates. For example, merely reversing the order of "Yes" and "No" nearly flips the model's retrieval tendency. Notably, RetrievalQA retrieves for all NaturalQA samples, contrary to the adaptive RAG objective. Figure 2 demonstrates that the keyword Shapley values are hardly distinguishable between answerable and unanswerable questions, and non-keywords present comparable influence to keywords.

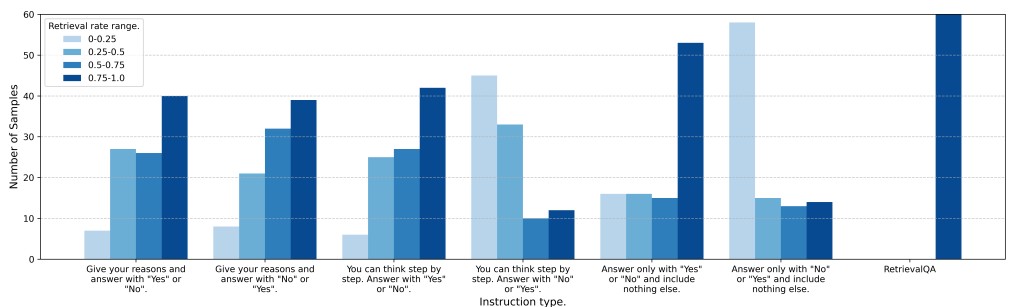

Figure 1: Retrieval rate distribution under different instructions.

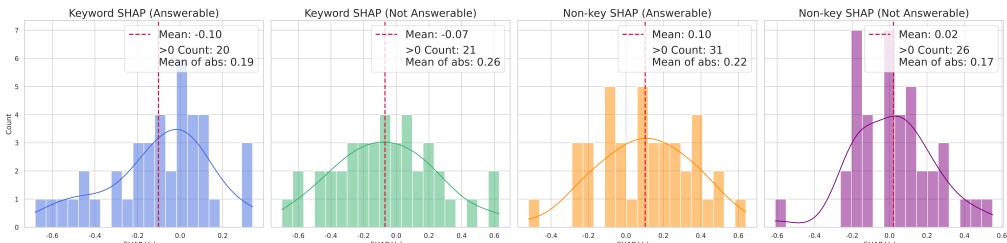

Figure 2: Shapley value analysis for keywords and non-keywords within queries.

These results suggest that prompting-based methods is highly dependent on superficial linguistic features and biases in instructions or queries, rather than genuinely assessing its own knowledge boundaries and make decisions accordingly. In fact, figure 2 indicates that the prompting-based retrieval decisions are not even statistically correlated to the actual necessity of retrieval. In this sense, existing prompting methods might be exploiting dataset-specific artifacts rather than reasoning about the required knowledge. For example, RetrievalQA benefits from the fact that much of the new-world and long-tail knowledge in QA datasets is missing from the LLM's internal knowledge, simply learning to fit the marginal distribution of retrieval rates.

**Experiment II: probing with a binary classifier.** Among the existing ARAG methods that probes into the LLM parameters, hidden states, and output probabilities, binary classifiers such as UAR (Cheng et al., 2024) have demonstrated strong in-distribution performance. UAR employs a single-layer MLP that maps hidden states to logits linearly, which makes its decision process more interpretable. In UAR, there is a classifier trained for time-sensitive queries (TAQA dataset, (Zhao et al., 2024)) in addition to one for general knowledge QA. We test whether the time-aware classifier can generalize to general knowledge QA.

| Similarity | T4T | T4N | N4N |
|---|---|---|---|
| T4T (TAQA, Time-aware model) | 1 | 0.9997 | 0.8893 |
| T4N (NaturalQA, Time-aware model) | | 1 | 0.8895 |
| N4N (NaturalQA, kNowledge-aware model) | | | 1 |
| Accuracy (%) | 76.58 | 40.58 | 58.94 |

Figure 3: Analysis of mechanistic similarity between different retrieval decision settings.

Figure 4: Comparing distributions of feature vector norms.

According to figure 3, while both classifiers do well in their corresponding domains, the time-aware classifier achieves only 40% accuracy on NaturalQA. Surprisingly, attribution analysis shows that

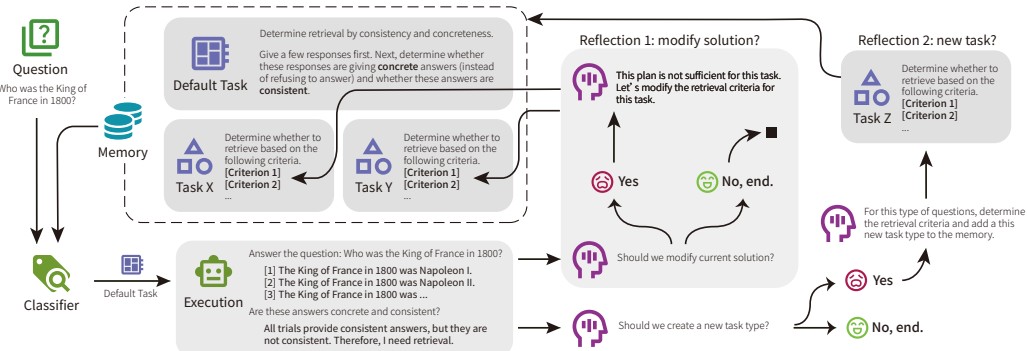

Figure 5: The overall framework structure for METAR.

these two domains share highly similar decision mechanisms (cosine similarity ≈ 0.9), which indicates that the failure arises from discrepancies in feature distribution. Indeed, we observe significant distributional shifts across domains for feature vector norms (in figure 4), one of the most low-dimensional statistics of the feature vector distribution. Such distributional shifts, combined with the probable failure of the decision mechanism itself in new domains and new tasks, restricts the applicability of such classifiers in real-world QA situations.

**Discussion** The experiments above have revealed that existing ARAG methods often fail at knowledge assessment and instead exploit task-related features or biases. Such features do not generalize: even when decision mechanism are highly similar, feature distribution shifts break the identically distributed assumption and cause collapse in retrieval decision performance. This highlights the importance of accurate knowledge boundary estimation, without which ARAG might degrade into simply fitting retrieval ratios for each task, preventing ARAG methods from improving efficiency and QA quality. Indeed, enabling LLMs to assess their own knowledge would reduce hallucinations fundamentally and allow safer LLM deployment in critical applications (e.g., autonomous driving).

## 4 METAR: TASK-AWARE ARAG WITH AN AGENTIC FRAMEWORK

In this section we present our agentic framework, **M**emory-**E**volving **T**ask-**A**ware **R**AG (METAR) for making retrieval decisions under various tasks and domains. As illustrated in Figure 5, METAR features a **memory** module and a **reflection** module specially designed for adaptive RAG. Unlike prior agentic frameworks focusing on reasoning or complex planning for tool-use, for example ReAct (Yao et al., 2023), AdaPlanner (Sun et al., 2023) and ToolChain (Zhuang et al., 2024), METAR makes use of its own capabilities to bootstrap its retrieval decision-making performance.

Noticeably, all components including decision-making, reflection and optimization are carried out in natural language without interaction with language mode parameters or hidden states. This makes the framework applicable to both white-box and black-box LLM settings. What's more, to ensure robustness in resource-constrained environments, such as using smaller open-source language models, our framework further emphasizes simplicity and clarity in design.

### 4.1 FRAMEWORK COMPONENTS AND WORKFLOW

We design the memory module as a *procedural memory*, which stores previously encountered QA tasks together with their corresponding retrieval solutions. This allows METAR to reuse past experience for new queries, thereby avoiding extensive reasoning and thus ensuring low decision latency when needed. The reflection module primarily operates on this memory. Rather than focusing solely on the current instance, the reflection module refines the stored solutions using new query samples.

During inference, given a query, the agent orchestrator first acts as a task classifier, determining whether the query matches an existing task in memory. If so, the corresponding solution is directly executed. Otherwise, the reflection module is invoked to propose a new task and its solution to

the memory module. It can also propose updates to the solution of an existing task if it considers the retrieval decision improper. Since the proposed modifications could damage the stability and effectiveness of the executor, they are only accepted after the orchestrator verifies their safety.

In figure 5 we show the detailed structure of the memory module. Each task solution is represented by a collection of evaluation criteria in JSON format. Such format enforcement effectively reduces the complexity of executing the solution, thereby increases reliability. In addition, we define a default task based on consistency judgments. This is motivated by the observation that most factual queries are handled fairly well by verifying **(1)** whether the LLM provides a concrete answer rather than avoid or refuse answering, and **(2)** whether it produces consistent answers across multiple attempts.

## 4.2 DISCUSSION

**What exactly is a task?** This question is inherently tricky but at the heart of the ARAG concept. Although explicit definitions of what constitutes a "task" are rarely provided, most methods are trained and evaluated on datasets that implicitly reflect different user goals. This suggests that task boundaries are ultimately determined by user intents. For example, NaturalQA targets general knowledge QA while RetrievalQA targets long-tail and new-world knowledge. But if we combine these two datasets, how should we define and describe the resulting task? The essential question is: *what is the intent behind mixing them?*

From this perspective, our orchestrator treats intent recognition as the primary goal in task classification for each query. We explicitly integrate intent recognition prompts into the orchestrator, requiring it to understand the purpose of the query, e.g., whether it is to search a historical event, retrieve recent news, or explore domain-specific knowledge. Thereby we ensure that the orchestrator can align each query with the most relevant user intent to guide retrieval decisions.

**Benefits of METAR** The design choice of METAR makes the retrieval criteria transparent and human-editable for real-world deployments. For example, when the system perform bad in some certain tasks, the human operator can directly look into the memory module and intervene by modifying it. Meanwhile, METAR allows flexible trade-offs. When latency is prioritized or computation is limited, the reflection module can be disabled. In this case, the system relies completely on the procedural memory without degrading performance, only sacrificing the benefits from continuous improvements.

## 5 EXPERIMENTS

In this section, we conduct experiments across various tasks to evaluate the task adaptability of existing ARAG methods and compare METAR against these baselines.

### 5.1 EXPERIMENT SETTINGS

**Baselines** We experiment with multiple baselines: **(1)** TAARE from RetrievalQA (Zhang et al. (2024)). This method optimizes its prompts for long-tail knowledge and new-world knowledge. **(2)** Self-RAG (Asai et al. (2023)). This is one of the most representative works in adaptive RAG using control tokens. It finetunes the language model on a large corpus of question answering data to make retrieval decisions. **(3)** UAR (Cheng et al. (2024)). In this work the authors train several classifiers aimed for several criteria of retrieval. A compiled list of ARAG methods is provided in the appendix section C.

**Tasks** We select a wide range of datasets serving as the set of tasks for adaptability evaluation. **(1)** NaturalQA (Kwiatkowski et al. (2019)) is a dataset derived from real user queries, and we would describe this task as generic factual questions. **(2)** RetrievalQA (Zhang et al. (2024)) comprises new-world and long-tail knowledge for short-form open-domain question answering. **(3)** SelfAware (Yin et al. (2023)) evaluates the LLMs' ability to recognize unaswerable questions, for example, subjective or philosophical ones. These questions do not have a definitive answer and retrieving external resources does little help. **(4)** TAQA (Zhao et al. (2024)) requires clear time-awareness to answer. The answers to these queries changes with time, and we reform the questions by specifying

a certain point in time, for example, "In 2010, who was the President of America?", to make these questions answerable. **(5)** UnknownBench (Liu et al. (2024)) are also unanswerable questions. This dataset is different from the unanswerable subset of SelfAware in that these queries are unanswerable due to erroneous premises or nonexistent concepts. Since the LLaMA-2 models can hardly recognize the false information in questions, we label all these samples as need external retrieval.

**Metrics**  For most question-answering datasets, the proportion of answerable questions by each backbone model can be unbalanced. Therefore, we use Matthews correlation coefficient (mcc) (Matthews (1975)) as a measure of binary classifications (equation 3) for most tasks. However, for the UnknownBench dataset where all samples are unanswerable, we simply use accuracy, and in adaptability calculation, we linearly map the accuracy metric to $[-1, 1]$.

$$\text{MCC} = \frac{\text{TP} \times \text{TN} + \text{FP} \times \text{FN}}{\sqrt{(\text{TP} + \text{FP})(\text{TP} + \text{FN})(\text{TN} + \text{FP})(\text{TN} + \text{FN})}}. \tag{3}$$

All the experiments are conducted on CPU: Intel(R) Xeon(R) Platinum 8336C CPU @ 2.30GHz and GPU: NVIDIA GeForce RTX 4090 using Python 3.9.21. We use LLaMA-2-13B-chat as our backbone model, the choice of which is because most of our baselines are implemented on the LLaMA-2 series of open-source LLMs.

## 5.2 RESULTS

The main results are presented in table 1. We can see that our default task performs fairly well for regular factual queries in NaturalQA, RetrievalQA and TAQA. This default workflow surpasses almost all existing ARAG methods in these factual queries except UAR, whose training corpus consists of TriviaQA (Joshi et al. (2017)), one of the data sources of RetrievalQA. When combined with other task memories, despite a minor drop in the performances of other tasks, we see that the framework successfully adapts itself to the unseen task SelfAware. As is demonstrated by Table 2, our framework is the only method that achieves high $P_{\mathcal{T}'}^{\text{worst}}$, ensuring an acceptable worst-case performance, and the expected performance $P_{\mathcal{T}'}^{\text{avg}}$ is comparable to the best existing method. These results indicate that our framework is an adaptable method applicable to various tasks.

Moreover, table 3 demonstrates the retrieval rates of existing RAG methods when applied to out-of-domain tasks. The uneven distributions between retrieval and not retrieval for most of these methods suggest that they might suffer from the feature distribution shift problem when encountered new tasks or new domains. This observation is aligned with the results of our empirical experiments in section 3.3, further providing validation of our Task-Dependent Knowledge Structure Hypothesis.

Table 1: Experiment results. A gray cell indicates that the method is optimized for this task.

| Method | NaturalQA (mcc) | RetrievalQA (mcc) | SelfAware (mcc) | TAQA (mcc) | UnknownBench (acc) |
|---|---|---|---|---|---|
| Baseline | -0.035 | -0.013 | -0.022 | -0.029 | 0.007 |
| Few-shot | -0.102 | -0.189 | -0.010 | 0.024 | 0.328 |
| TAARE | 0.016 | 0.042 | 0.228 | 0.013 | **0.912** |
| Self-RAG | -0.022 | -0.015 | 0.080 | 0.076 | 0.415 |
| Self-RAG(t=0.5) | -0.091 | 0.013 | -0.195 | 0.052 | 0.066 |
| UAR | 0.139 | **0.478** | 0.376 | 0.050 | 0.856 |
| Default Task | **0.225** | 0.427 | 0.073 | **0.204** | 0.744 |
| METAR | 0.195 | 0.362 | 0.181 | 0.164 | 0.754 |

Table 2: Task adaptability measures.

| Measure | Baseline | Few-shot | TAARE | Self-RAG | Self-RAG(t=0.5) | UAR | Default Task | METAR |
|---|---|---|---|---|---|---|---|---|
| $P_{\mathcal{T}_{\text{remain}}}^{\text{avg}}$ | -0.217 | -0.124 | 0.270 | -0.005 | -0.337 | 0.319 | **0.3284** | 0.282 |
| $P_{\mathcal{T}_{\text{remain}}}^{\text{worst}}$ | -0.986 | -0.344 | 0.013 | -0.170 | -0.868 | 0.05 | 0.073 | **0.164** |

Table 3: Retrieval rates.

| Method | NaturalQA | RetrievalQA | SelfAware | TAQA | UnknownBench |
|---|---|---|---|---|---|
| Baseline | 0.014 | 0.022 | 0.063 | 0.001 | 0.007 |
| Few-shot | 0.453 | 0.346 | 0.434 | 0.365 | 0.328 |
| TAARE | 0.941 | 0.954 | 0.841 | 0.986 | 0.912 |
| Self-RAG | 0.001 | 0.001 | 0.524 | 0.354 | 0.415 |
| Self-RAG(t=0.5) | 0.156 | 0.237 | 0.314 | 0.046 | 0.066 |
| UAR | 0.139 | 0.654 | 0.348 | 0.900 | 0.856 |
| Default Task | 0.313 | 0.501 | 0.281 | 0.600 | 0.744 |
| Framework | 0.401 | 0.551 | 0.309 | 0.668 | 0.754 |

## 6 CONCLUSION

In this paper, we propose the task adaptability problem in adaptive RAG settings and provide intuitive explanations to understand this problem. We benchmark existing ARAG works to validate that current methods generally lacks adaptability to diverse or unseen tasks. To tackle the adaptability problem, we propose METAR, a novel agentic framework that stores and refines its solutions through carefully designed memory module and reflection module. Our experiments show that our framework outperforms most existing ARAG methods in making retrieval decisions and is the first ARAG framework that is adaptable to a variety of tasks.

However, there are also limitations of this study. We summarize the limitations of our work as follows. First, our designed framework achieves task adaptability at the cost of breaking down the decision process and thus increasing inference latency. This increase in latency is mainly because we have bypassed the difficulty in self-knowledge awareness instead of directly trying to overcome such difficulty. We deem it a promising area for further investigation. Next, the framework primarily focus on the short-form question-answering tasks. Tasks that involve long reasoning over factual evidences to derive an answer might not be applied to this framework.

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

## A  LLM USAGE

LLMs are used to aid the writing and polishing of this manuscript. Specifically, we use an LLM to assist in refining the expression, e.g., sentence rephrasing and grammar checking, (mainly) in section 3 and 4 in order to increase readability. However, LLMs are not involved in research methodology or experimental designs. All the concepts, ideas, and analyses are developed solely by the authors. We take full responsibility for the content of the manuscript, including any text generated or polished by the LLM. We have ensured that the LLM-generated text adheres to ethical guidelines and does not contribute to plagiarism or scientific misconduct.

# B   DETAILS OF THE EMPIRICAL STUDIES

For the experiments in section 3.3, we use LLaMA-2-7b-chat as the backbone model in consideration of the computational .

## B.1   DETAILS OF THE SHAPLEY VALUES ANALYSIS

The Shapley value (Shapley, 1951) is a concept from cooperative game theory which provides a method to allocate the payoff among players by evaluating their contribution to the utility of the whole group. Mathematically, given a set of players $P$ and a value function $v : 2^P \rightarrow \mathbb{R}$ that assigns a utility to each subset of players, the Shapley value of any player $i \in P$ is calculated as

$$\phi_i(v) = \sum_{S \subseteq P\{i\}} \frac{|S|!(|P| - |S| - 1)!}{|P|!} \Big( v\big(S \cup \{i\}\big) - v\big(S\big) \Big) \tag{4}$$

In the mechanistic interpretability field of machine learning, the Shapley value has been adopted widely, especially by the works that strives to model the input features of a LLM to a low-dimensional output (e.g., a retrieval decision). Notably, this method involves exponential calculations of the value function $v$, so the computational burden when applied to high-dimensional inputs grows also exponentially. In order to attribute the keyword tokens in a QA query to the retrieval decision, an ideal implementation would be treating each token as an independent feature and then sum up the Shapley values of keyword tokens, but this setting leads to unacceptably high LLM calls. Therefore, we compromise and take keyword and non-keyword tokens as only two independent features.

## B.2   IMPLEMENTATION OF EXPERIMENT I (UNDERSTANDING PROMPTING METHODS).

Specifically, for the 100 samples from NaturalQA (50 answerable, 50 unanswerable), we calculate the retrieval rates by doing roll-out under six near-equivalent instructions. The number of roll-outs is selected as 50, in a trade between estimation accuracy and computational resource consumption. The instructions consists of "Determine whether you need to retrieve external resources to answer the following question." and different formatting instructions provided in figure 1. It leads to semantically identical instructions that differ only in phrasing details like swapping the order of "Yes/No" or including an explicit chain-of-thought prompt.

As we have described in section 3.3, we categorize query tokens into keywords and non-keywords, yielding two features groups for the Shapley value analysis. We annotate the keywords in each query with a ChatGPT API call, simulating human annotations. As an example, in the query "Who was the King of France in 1800?", keywords include "King of France" and "1800", while the other tokens are considered non-keywords. Keywords or non-keywords are replaced with the unknown token ("<unk>" for Llama-2 models) to perform mean-ablation in Shapley value calculation. This special token is chosen because the LLMs generally tries to ignore the semantics of the token while being aware that there exists some missing features.

## B.3   IMPLEMENTATION AND SUPPLEMENTARY RESULTS OF EXPERIMENT II (ANALYSIS OF UAR).

As for the analysis of UAR classifiers, we test whether the time-aware classifier generalizes to general knowledge QA rather than the reverse. This design choice is because TAQA dataset has a clearer temporal distribution, while general QA datasets are more heterogeneous and may include time-sensitive questions, introducing confounding factors into the transfer evaluation.

The UAR classifiers works as follws: given input features $\mathbf{h}$, two logits $\boldsymbol{W}_1 \mathbf{h}$ and $\boldsymbol{W}_0 \mathbf{h}$ are computed (linearly) for retrieval versus non-retrieval. Then a softmax function normalizes the logits, yielding decision probability $(P', P) = \text{softmax}(\boldsymbol{W}_0 \mathbf{h}, \boldsymbol{W}_1 \mathbf{h})$. To compare the decision mechanisms of classifiers across domains, we compute the gradients of the decision probability with respect to input features $\partial P / \partial \mathbf{h}$ and take the cosine similarity between gradient vectors as the measure of mechanistic similarity (figure 3). The three gradient vectors are visualized as in figure 6. Meanwhile, we plot the distributions of projection scores (which reflect the separation between positive and negative samples) in figure 7. The projection score visualization indicates that classifiers struggle to distinguish positive

and negative samples even within their own domains, which means the prediction by these classifiers do not exhibit statistical correlations with the actual retrieval necessity.

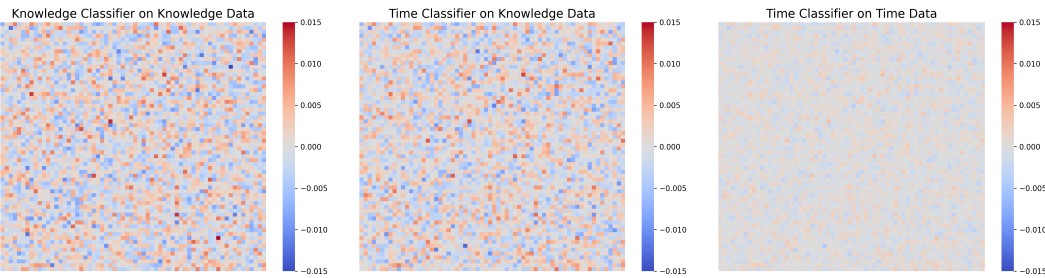

Figure 6: The decision mechanism visualization.

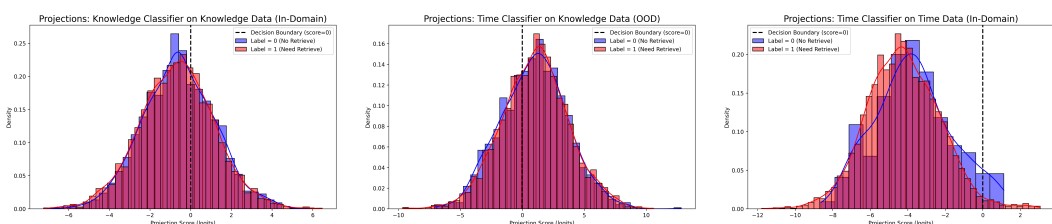

Figure 7: The distributions of projection score, which corresponds to the decision logits.

## C  LIST OF EXISTING METHODS

In table 4 we compile existing representatives of ARAG methods together with their task or domain that is suitable to apply to.

Table 4: Existing methods for adaptive RAG.

| Method (Category) | Task Type | Data Source |
|---|---|---|
| RetrievalQA (prompt) | New-world Knowledge, Long-tail Knowledge | RealTimeQA, FreshQA PopQA, TriviaQA, ToolQA |
| Self-RAG (control token) | Open-domain QA | NaturalQA, TriviaQA, PopQA, ASQA, OBQA, KILT |
| DRAGIN (Thresholding) | Multi-hop QA, Reading Comprehension, Commonsense Reasoning | HotpotQA, 2WikiMultihopQA IIRC StrategyQA |
| ROWEN (Thresholding) | Factual QA, Commonsense Reasoning | TriviaQA StrategyQA |
| UAR (Probing) | Time-sensitive Knowledge, Factual QA | TAQA TriviaQA |
| ProbingRAG (Probing) | Factual QA, Multi-hop QA | NaturalQA, TriviaQA HotpotQA |

