# OpenReview forum: "Beyond Static Retrieval Policies: Task-Aware Adaptive RAG With METAR"
_ICLR.cc/2026/Conference — Submitted to ICLR 2026_

### Official Review · Reviewer_oTmg · 2025-10-18

**Soundness:** 2
**Presentation:** 2
**Contribution:** 2
**Rating:** 2
**Confidence:** 3

**Summary:**

This paper discusses the issue in Adaptive Retrieval-Augmented Generation, the task adaptability of retrieval decision policies.
The authors formalize the "task adaptability" problem and propose the Task-Dependent Knowledge Structure Hypothesis, arguing that LLMs encode knowledge differently across domains, which may make static retrieval policies brittle. They then propose METAR that uses a procedural memory and a reflection mechanism to evolve task-specific retrieval criteria.

**Strengths:**

- Clear framework structure and motivation
- Consistent results across QA tasks.

**Weaknesses:**

- The "procedural memory" is mentioned as a list of task-specific criteria, but there is no formal definition of how tasks are recognized, merged, or pruned. The task is vague, even in the discussion section.

- The misleading result. The work proposed the framework METAR. But no evidence how this framework addresses the knowledge heterogeneity problem for retrieval?

- There is no ablation showing the separate contributions of the task memory module, reflection module, or the default consistency heuristic. How does this so-called "task-specific criteria" improve the performance?

- Table 1 and 2 show that METAR’s improvements are small and occasionally worse on seen tasks.

- Since METAR involves multiple reasoning and reflection steps, the inference cost could be significantly higher, but the paper provides no runtime or latency analysis.

**Questions:**

How is task recognition implemented in the orchestrator? Is it prompt-based classification, clustering, or heuristic matching?

How is METAR solve this heterogeneous knowledge problem?

---

### Official Review · Reviewer_e5U8 · 2025-10-24

**Soundness:** 3
**Presentation:** 1
**Contribution:** 2
**Rating:** 2
**Confidence:** 4

**Summary:**

The paper proposes a new framework called METAR (Memory-Evolving Task-Aware RAG) aimed at enhancing the adaptability of Retrieval-Augmented Generation (RAG) systems. The authors identify a key problem in current adaptive retrieval methods: their lack of adaptability when applied to tasks that differ from those seen during training. To address this, METAR introduces a procedural memory and reflection mechanism to dynamically adjust retrieval policies based on task requirements. The authors present experimental results comparing METAR with existing adaptive retrieval methods, demonstrating its advantages in terms of adaptability across diverse tasks.

**Strengths:**

1. The paper identifies a crucial gap in current ARAG approaches—task adaptability. While existing methods often perform well for specific tasks, they struggle with transferring to new, unseen tasks. This gap has been well recognized and is a relevant issue in the deployment of ARAG in real-world applications.
2. METAR introduces the idea of a procedural memory system that evolves based on feedback, which could provide a more flexible and adaptable retrieval decision mechanism. This is an interesting contribution and could lead to future research in task-aware retrieval systems.

**Weaknesses:**

1. While the idea of task-aware retrieval is valuable, the novelty of METAR is limited. The concept of memory-based frameworks for improving task adaptability in AI systems has been explored before in other contexts, including multi-task learning and lifelong learning. The contribution does not appear sufficiently groundbreaking, especially when compared to similar agentic frameworks like ExpeL, Memp or AutoGPT. The novelty lies more in the specific combination of components rather than introducing a radically new concept.
2.  The framework is evaluated on a limited set of tasks, which lack of an evolved scenario to demonstrate the adaptation process over time. While METAR is designed to adapt based on task-specific feedback through its procedural memory, the paper does not present a detailed, step-by-step evolution of how the framework improves and refines its retrieval policies as it encounters new tasks.
3. The paper could benefit from clearer presentation and structure. The discussion of METAR’s components (memory, reflection) is somewhat vague, and the workflow diagram (Figure 5) does not provide enough clarity about how the different modules interact. A more detailed, step-by-step explanation of the decision-making process in METAR is needed.

**Questions:**

See Weaknesses.

---

### Official Review · Reviewer_uALD · 2025-10-28

**Soundness:** 2
**Presentation:** 2
**Contribution:** 2
**Rating:** 4
**Confidence:** 4

**Summary:**

This paper formally identifies the adaptability issue inherent in existing Adaptive Retrieval-Augmented Generation (ARAG) techniques for the first time. The authors propose a novel matrix to benchmark the adaptability of current ARAG methods. The paper also put forward a hypothesis to analyze the underlying cause of this phenomenon, attributing it to the distributional shifts of features across different tasks. Furthermore, the paper proposes METAR as a solution to this problem and validates its effectiveness through experiments.

**Strengths:**

1. The paper is the first to identify the adaptability problem in ARAG systems and designs a novel framework METAR to address it, which is training-free and natural-language-based.

2. The paper empirically validates the prevalence of the adaptability issue in existing ARAG methods and demonstrates the effectiveness of the proposed METAR framework.

3. The METAR framework is modular and can potentially incorporate other ARAG methods as "solutions" within its procedural memory.

**Weaknesses:**

1. The proposed solution appears to be more engineering-oriented and lacks significant algorithmic novelty.

2. The paper mentions disabling the reflection module in resource-constrained scenarios. In such cases, The system's performance may degrade when encountering new tasks.

3. The paper lacks ablation studies comparing the memory module and the reflection module, making it difficult to assess the individual contribution of each component to the overall results.

4. Experimental results show that using the Default Task alone yields good performance, while storing other tasks sometimes leads to performance degradation. Does this suggest that the memory module might impair adaptability in some cases?

5. The agentic workflow, particularly the reflection and memory operations, likely increases inference latency significantly compared to static methods. This could limit the applicability of METAR in real-time or interactive scenarios.

6. There are some formula and word spelling errors in the paper which need correction, for instance, in Equation 3.

**Questions:**

1. How are the criteria for storing task memories established? On what basis are they defined?

2. How does the METAR framework handle memory growth and potential conflicts as the number of stored tasks scales to dozens or hundreds?

3. What safeguards are in place to prevent it from making erroneous updates to the procedural memory that may degrade performance?

---

### Official Review · Reviewer_diJq · 2025-10-31

**Soundness:** 1
**Presentation:** 2
**Contribution:** 1
**Rating:** 2
**Confidence:** 4

**Summary:**

The paper claims that existing active retrieval methods exhibit poor cross-task generalization, as they are primarily based on prompt designs or shallow classifiers that perform well only within a single task but degrade significantly when applied to new ones. To address this limitation, the authors propose a new framework, METAR, which employs a “memory–reflection” mechanism, enabling the model to adapt its retrieval strategies dynamically across different tasks rather than relying on fixed rules.

**Strengths:**

1. The paper defines the problem of task adaptability to study the cross-task generalization of active retrieval methods, which adds a degree of novelty.

2. The writing is clear and well-organized, and the figures are visually appealing, contributing to strong overall presentation quality.

**Weaknesses:**

1. The experimental scope is narrow and small-scale — it only covers five QA datasets with limited evaluation, without deeper analysis of reasoning latency or efficiency.

2. The proposed method shows limited novelty — the “Memory-Evolving” module essentially records retrieval rules in natural language (JSON format) and modifies them through a reflection mechanism.

**Questions:**

1. The authors should further clarify the practical significance of the “whether to retrieve” task formulation. Recent active retrieval research (e.g., Search-o1, Search-R1, AstuteRAG) focuses on more complex problems such as when to retrieve, what to retrieve, and how to perform multi-round retrieval and evidence integration. Compared with these works, does the formulation in this paper oversimplify the problem and fail to capture the real reasoning–retrieval process?

2. The novelty of the proposed Memory-Evolving module needs further clarification. Is this module essentially just storing and updating task–retrieval rules in JSON format, without introducing a fundamentally new algorithmic mechanism?

3. The paper claims to be the first to define task adaptability, but it remains unclear how this concept differs from existing notions such as domain generalization or task transfer. The theoretical distinction should be clarified.

4. The paper lacks systematic ablation studies to verify the independent contribution of each component (e.g., memory, reflection, and rule adaptation). Moreover, the experiments are conducted on only a few QA datasets and do not cover multi-hop reasoning or generation-oriented tasks, making it difficult to demonstrate the general effectiveness of the proposed method.

---

### Meta-Review · Area_Chair_kfHx · 2026-01-07

**Summary:**

The paper proposes METAR (Memory-Evolving Task-Aware RAG), a framework designed to improve the cross-task adaptability of active retrieval systems by employing a memory-reflection mechanism. While the problem of task adaptability in RAG is recognized as valuable, reviewers unanimously criticized the method for limited novelty, lack of rigorous evaluation, and incremental engineering. The consensus is that the proposed solution is essentially a prompt-based rule recorder without algorithmic depth, and the experimental validation is insufficient to support the claims of superior adaptability.

**Reviewer Concerns:**

Limited Novelty (Reviewers diJq, e5U8, uALD): The core mechanism is viewed as a trivial application of memory-based agents (storing rules in JSON) rather than a novel algorithm. It lacks differentiation from existing agent frameworks like AutoGPT or ExpeL.

Insufficient Evaluation (Reviewers diJq, oTmg, uALD): The experiments are small-scale (only 5 QA datasets), lack ablation studies to isolate component contributions, and do not analyze latency or cost, which are critical for agentic workflows.

Vague Methodology (Reviewers oTmg, e5U8): Key processes like task recognition, memory pruning, and rule evolution are not formally defined. The "procedural memory" appears to be a loose collection of heuristics without a robust update mechanism.

Marginal Gains (Reviewer oTmg): Empirical improvements are small or inconsistent, sometimes even degrading performance on seen tasks.

**Reviewer Scores:**

Reviewer diJq: 2 (Reject)

Reviewer uALD: 4 (Borderline Reject)

Reviewer e5U8: 2 (Reject)

Reviewer oTmg: 2 (Reject)

---

### Decision · Program_Chairs · 2026-01-26

Reject